# Longitudinal host transcriptional responses to SARS-CoV-2 infection in adults with extremely high viral load

Vasanthi Avadhanula[1], Chad J. Creighton[2,3,4]*, Laura Ferlic-Stark[1], Divya Nagaraj[1], Yiqun Zhang[2], Richard Sucgang[5], Erin G. Nicholson[1,6], Anubama Rajan[1], Vipin Kumar Menon[4,7], Harshavardhan Doddapaneni[4,7], Donna Marie Muzny[4,7], Ginger A. Metcalf[4,7], Sara Joan Javornik Cregeen[1], Kristi Louise Hoffman[1], Richard A. Gibbs[4,7], Joseph F. Petrosino[1], Pedro A. Piedra[1,6]*

1 Department of Molecular Virology and Microbiology, Baylor College of Medicine, Houston, TX, United States of America, 2 Dan L. Duncan Comprehensive Cancer Center Division of Biostatistics, Baylor College of Medicine, Houston, TX, United States of America, 3 Department of Medicine, Baylor College of Medicine, Houston, Texas, United States of America, 4 Human Genome Sequencing Center, Baylor College of Medicine, Houston, Texas, United States of America, 5 Houston Methodist Research Institute, Center for Health Data Science and Analytics, Houston, Texas, United States of America, 6 Department of Pediatrics, Baylor College of Medicine, Houston, Texas, United States of America, 7 Department of Molecular and Human Genetics, Baylor College of Medicine, Houston, Texas, United States of America

☯ These authors contributed equally to this work.
* ppiedra@bcm.edu (PAP); creighto@bcm.edu (CJC)

**Data Availability Statement:** The RNA-seq dataset of serially collected samples and of nose organoids have been deposited at Gene Expression Omnibus (GEO) (GEO accession number- GSE234487

## Abstract

Current understanding of viral dynamics of SARS-CoV-2 and host responses driving the pathogenic mechanisms in COVID-19 is rapidly evolving. Here, we conducted a longitudinal study to investigate gene expression patterns during acute SARS-CoV-2 illness. Cases included SARS-CoV-2 infected individuals with extremely high viral loads early in their illness, individuals having low SARS-CoV-2 viral loads early in their infection, and individuals testing negative for SARS-CoV-2. We could identify widespread transcriptional host responses to SARS-CoV-2 infection that were initially most strongly manifested in patients with extremely high initial viral loads, then attenuating within the patient over time as viral loads decreased. Genes correlated with SARS-CoV-2 viral load over time were similarly differentially expressed across independent datasets of SARS-CoV-2 infected lung and upper airway cells, from both in vitro systems and patient samples. We also generated expression data on the human nose organoid model during SARS-CoV-2 infection. The human nose organoid-generated host transcriptional response captured many aspects of responses observed in the above patient samples, while suggesting the existence of distinct host responses to SARS-CoV-2 depending on the cellular context, involving both epithelial and cellular immune responses. Our findings provide a catalog of SARS-CoV-2 host response genes changing over time and magnitude of these host responses were significantly correlated to viral load.

**Funding:** This work was supported by two NIH grants: CA125123 (CJC) and U19AI144297 (VA, CJC, RG, JP, PAP).

**Competing interests:** The authors have declared that no competing interests exist.

**Abbreviations:** COVID-19, Coronavirus disease 19; SARS-CoV-2, severe acute respiratory syndrome coronavirus 2.

## Introduction

Severe acute respiratory syndrome coronavirus-2 (SARS-CoV-2) is the etiologic agent of the coronavirus disease 2019 (COVID-19) pandemic. The clinical spectrum of COVID-19, caused by SARS-CoV-2, is wide, ranging from asymptomatic infection to fatal disease. Risk factors for severe illness and death include age, sex, smoking, and comorbidities, such as obesity, hypertension, diabetes, and cardiovascular disease. Studies suggested that SARS-CoV-2 viral load can predict the likelihood of disease spread and severity [1–3]. A higher detectable SARS-CoV-2 plasma viral load was associated with worse respiratory disease severity [4]. Conversely, robust immune responses putatively mediate non-severe illness, in part, by controlling the replication of SARS-CoV-2 [5,6]. Emerging evidence indicates that age and sex differences in the innate and adaptive immune response can explain the higher risks observed in older adults and male cases [7,8].

Initial site of SARS-CoV-2 replication is the upper respiratory tract, and replication usually peaks within the first week of infection [6]. The amount of virus produced at the respiratory epithelium is considered to be a critical element in determining SARS-CoV-2 transmissibility, duration of illness or severity, although it is not the only factor [9,10]. Higher viral loads have been observed in hospitalized patients with severe disease, have been attributed to high transmission and superspreading events, and have resulted in prolonged viral RNA shedding [1,11–15].

Specific anatomic site or host cell type where viral replication occurs, can also determine the course of infection. For example, angiotensin-converting enzyme 2 (ACE-2) and trans-membrane serine protease 2 (TMPRSS2) receptors expression is highest in the upper respiratory tract and decreases in the distal or lower respiratory tract, incidentally SARS-CoV-2 infection mirrored this pattern, with high replication in proximal (nasal) versus distal pulmonary (alveolar) epithelial cells [16]. Control of viral replication and resolution of the inflammatory response is believed to be dependent, in part, on viral load and route of infection as well as the host immune response [17]. The early host immune response is regulated closely by the epithelial cell cytokine signaling in response to active viral replication [18]. Rapid and robust activation of the antiviral innate immune response at the site of viral replication is required to control and clear the virus. A delayed cytokine response can result in prolonged viral replication and worst clinical outcome as seen for other respiratory viruses [19].

Our understanding of the viral dynamics of SARS-CoV-2 and host responses driving the pathogenic mechanisms in COVID-19 is evolving rapidly. Multiple studies have reported various characteristics of immune/inflammatory responses to SARS-CoV-2. Cytokine or chemokines-related host inflammatory responses such as CCL2/MCP-1, CXCL10/IP-10, CCL3/MIP-1A, and CCL4/MIP-1B were detected in bronchoalveolar lavage samples of SARS-CoV-2 infected adults while activation of apoptosis and the P53 signaling pathway were observed in lymphocytes [20]. Inflammatory cytokine such as IL-1, IL-18, and IL-33 were enriched in the airways of COVID-19 patients [21]. In addition, a shotgun host transcriptomic analysis on nasopharyngeal samples revealed a wide range of antiviral responses. These included gamma and alpha interferon responses, elevated levels of ACE-2, interferon stimulated genes (ISGs), and interferon inducible (IFI) genes [22]. Very few studies have demonstrated the temporal correlation between viral load and host gene expression. Variation in viral load was associated with the SARS-CoV-2 disease and the host response dynamics via innate and adaptive immunity[23]. Another study revealed that expression of interferon-responsive genes, including ACE-2, increased as a function of viral load, while transcripts for B cell–specific proteins and neutrophil chemokines were elevated in patients with lower viral load [24]. Rouchka et.al. reported that cellular antiviral responses strongly correlated with viral loads. Though, COVID-

19 patients who experienced mild symptoms had a higher viral load than those with severe complications [6].

We previously reported on a small group of adults with extremely high SARS-CoV-2 viral load, who had the potential to be super spreaders and a large group of adults with low SARS-CoV-2 viral load, both groups had mild illness [14]. Here, we wanted to determine the host response in relation to the viral load early during infection and how it changes over time. We conducted a longitudinal study to investigate gene expression patterns detected in the secretion of the nasal epithelium during the acute phase of SARS-CoV-2 infection. The cases included SARS-CoV-2 infected individuals with an extremely high viral load early in their illness matched to individuals who either had a low SARS-CoV-2 viral load early in their infection or were otherwise stable patients who tested negative for SARS-CoV-2 prior to their outpatient surgical or aerosol generating procedure. We also determined the transcriptional response of a human nose organoid (HNO) line infected with SARS-CoV-2 and compared it to transcriptomic profiles generated from the upper respiratory tract secretion collected by nasal swabs from SARS-CoV-2 infected individuals.

## Results

### RNA sequencing of serially collected specimens

The demographic and visit characteristics for the selected cohort is presented in S1 Table. In general, age, gender, race, ethnicity, and zip code were comparable between the extremely high viral load, low viral load, and SARS-CoV-2 negative adults. Of the 73 MT swab samples from the extremely high and low viral load SARS-CoV-2 groups with longitudinal follow-up and SARS-CoV-2 negative controls, only 44 (60.3%) MT swab samples from 20 (66.7%) individuals were of good quality to generate RNA-sequence data to study the host response to SARS-CoV-2 infection over time. Demographic factors such as age, gender, race, ethnicity, zip code, disease severity and co-morbid conditions were comparable between the extremely high viral load, low viral load groups, and SARS-CoV-2 negative control group (Table 1).

### Gene expression changes by viral load

From our RNA-seq dataset, we could identify widespread gene expression changes from the nasal epithelium attributable to transcriptional host responses to SARS-CoV-2 infection. By comparing the expression levels of each gene with the sample viral load (representing the inverse correlation with Ct value) across the 44 MT swab samples, 429 genes were statistically correlated at $p<0.01$ significance level and 112 genes at $p<0.001$ (Fig 1A, Pearson's correlation). A stricter statistical cutoff would involve fewer expected false positive genes from multiple testing. However, the above 429 genes with $p<0.01$ would still be highly enriched for true positives, as revealed by integrating these genes with information from external databases, as described below. We also compared the expression levels of genes at individual time points during infection of both the extremely high viral load and low viral load groups with the SARS-CoV-2 negative control group (Fig 1B). Comparing Visit 1 MT swab samples from the extremely high viral load cases (n = 8 samples from eight subjects) with the MT swab samples in the SARS-CoV-2 negative control group (n = 4) yielded the highest number of genes with statistically significant correlated expression, as opposed to comparisons involving later times for the extremely high viral load group or involving the low viral load group. The gene expression from the extremely high viral load cases at Visit 1 highly overlapped with the differentially expressed genes of the low viral load group at Visit 1 (Fig 1C) and remained highly correlated throughout their last visit. Interestingly, genes from the extremely high viral load group that

**Table 1. Demographic characteristics of the matched groups with good quality RNA sequencing data.**

| | Extremely high viral load cases (N = 8) | Low viral load cases (N = 8) | Negative for SARS-CoV-2 (N = 4) | p-value |
|---|---|---|---|---|
| **Age** | | | | 0.879[a] |
| Median (Q1, Q3) | 39.5 (27.5, 57.5) | 46.0 (33.0, 55.0) | 45.5 (32.0, 57.0) | |
| Min, Max | 22.0, 60.0 | 24.0, 80.0 | 30.0, 57.0 | |
| **Gender** | | | | 0.851[b] |
| Male | 4 (50.0%) | 5 (62.5%) | 3 (75.0%) | |
| Female | 4 (50.0%) | 3 (37.5%) | 1 (25.0%) | |
| **Race** | | | | 0.158[b] |
| Asian | 1 (12.5%) | 1 (12.5%) | 0 (0.0%) | |
| Black | 2 (25.0%) | 0 (0.0%) | 0 (0.0%) | |
| White | 4 (50.0%) | 2 (25.0%) | 1 (25.0%) | |
| Other/Multiracial | 1 (12.5%) | 1 (12.5%) | 0 (0.0%) | |
| Unknown | 0 (0.0%) | 4 (50.0%) | 3 (75.0%) | |
| **Ethnicity** | | | | 0.297[b] |
| Hispanic | 2 (25.0%) | 3 (37.5%) | 1 (25.0%) | |
| Non-Hispanic | 6 (75.0%) | 3 (37.5%) | 1 (25.0%) | |
| Unknown | 0 (0.0%) | 2 (25.0%) | 2 (50.0%) | |
| **Disease Severity** | | | | 0.603[b] |
| Asymptomatic/Mild | 3 (37.5%) | 3 (37.5%) | 3 (75.0%) | |
| Mild/Moderate | 5 (62.5%) | 5 (62.5%) | 1 (25.0%) | |
| **Number of Co-morbid Conditions** | | | | 0.656[b] |
| None | 5 (62.5%) | 4 (50.0%) | 4 (100.0%) | |
| One | 1 (12.5%) | 3 (37.5%) | 0 (0.0%) | |
| Two | 1 (12.5%) | 0 (0.0%) | 0 (0.0%) | |
| Three + | 1 (12.5%) | 1 (12.5%) | 0 (0.0%) | |
| **Sample Collected** | | | | |
| at Visit 1 only | 0 | 1 | 4 | |
| at Visits 1, 2, 3 | 5 | 2 | 0 | |
| at Visits 1, 2, 3, 4 | 1 | 0 | 0 | |
| at Visit 2 only | 0 | 2 | 0 | |
| at Visits 1, 2 | 0 | 1 | 0 | |
| at Visits 1, 3 | 2 | 0 | 0 | |
| at Visits 1, 2, 4 | 0 | 1 | 0 | |
| at Visits 1, 3, 4 | 0 | 1 | 0 | |
| **Number of Samples per Subject** | | | | |
| One | 0 | 3 | 4 | |
| Two | 2 | 1 | 0 | |
| Three | 5 | 4 | 0 | |
| Four | 1 | 0 | 0 | |

Differences between groups were determined using the Kruskal-Wallis test for variables with non-parametric distribution and by Fisher's Exact test for categorical variables. P-value <0.05 was considered significantly different between groups.

did not overlap with the low viral load group did not show significant overlap with analogous differential patterns from external SARS-CoV-2 databases.

To further delineate the differences in host gene expression between extremely high and low SARS-CoV-2 viral load groups, we performed an UpSet plot analysis to identify unique

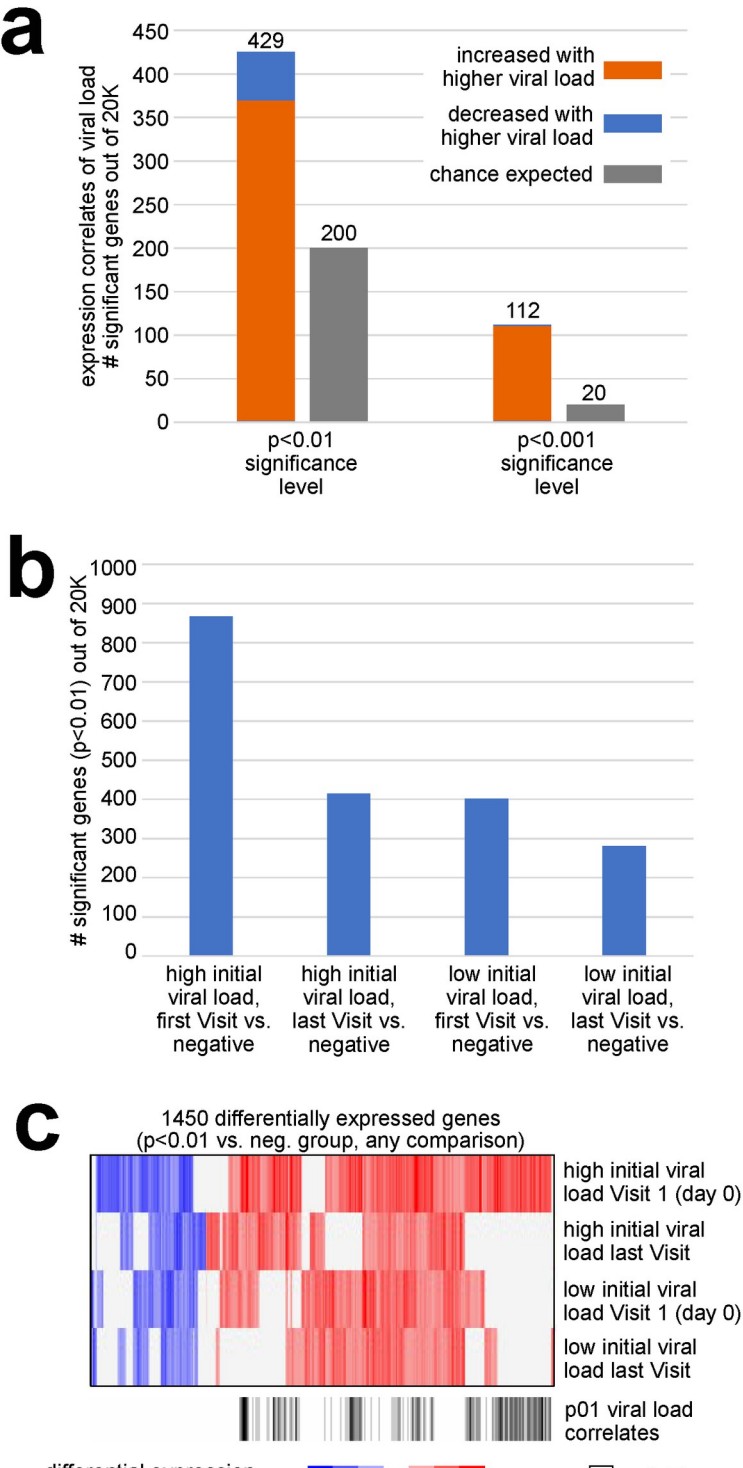

**Fig 1. Differential gene sets associated with the transcriptional host response to SARS-CoV-2 infection across serially collected samples. (a)** For each of 20000 genes, expression (log2 FPKM) was correlated with viral load (inverse correlation with Ct value) across 44 samples from 20 subjects. Numbers of statistically significant genes (by Pearson's) at both p<0.01 and p<0.001 significance levels are represented, as compared to the chance expected by multiple testing [25]. **(b)** Numbers of differential genes (p<0.01, t-test) when comparing: 1) Visit 1 samples from the extremely high viral load group (n = 8 samples from eight subjects) with the samples in the negative group (n = 4); samples at the

latest time points for each of the subjects from the extremely high viral load group (n = 8 samples) with the samples in the negative group; samples from the low viral load group (n = 8 samples from eight subjects, using earliest time point) with the samples in the negative group. Chance expected genes at p<0.01 due to multiple testing would be on the order of 200 [25]. **(c)** Heat map comparing differential patterns across the three comparisons from part b, for the 1357 genes significant (p<0.01) for any comparison. Row along the bottom indicated which genes were positively correlated with viral load (p<0.01) across all 44 samples (from part a), and which genes have Gene Ontology (GO) annotation [26] 'response to virus'. Visualization using heat maps was performed using JavaTreeview (version 1.1.6r4) [27].

and common intersecting genes between the samples (Fig 2A). Among all the differentially expressed genes (DEGs) in the samples, 614 DEGs were unique to the subjects in the extremely high viral load group at visit 1 (first visit) and 226 genes were unique for the extremely high viral load at the last visit. The low viral load subjects on the first and last visit showed 157 and 93 unique DEGs respectively. There were 31 DEGS that were common between all the groups. We performed the Gene ontology (GO) analysis of the unique and overlapping DEGs sets, and we found significant enrichment (FDR <0.05, count = 3) of the biological processes including defense response to virus, negative regulation of viral genome replication, innate immune response, response to virus (Fig 2B) that were uniquely expressed in the extremely high viral load group at visit 1. SARS-CoV-2 infection in the low viral load group at either the early or later phase of the infection and the extremely high viral load group at the last visit did not show statistically significant enrichment of GO biological process. These findings indicate that subjects with extremely high viral load at their initial visit were responding to the infection with increased immune responses, and thus preventing prolonged viral infection with a poor prognosis.

## Differentially expressed genes in respiratory samples from extremely high viral load adults

Focusing on the 112 top gene expression correlates of viral load across the 44 MT swab samples (p<0.001, Pearson's), 108 of these genes were higher in the SARS-CoV-2 infected adults with extremely high viral load. When visualizing the differential expression patterns of these 108 genes by heat map (Fig 3A), the genes were highest at Visit 1 of the extremely high viral load group, then decreased in expression with subsequent time points, tracking with the decrease in viral load (i.e., increase in Ct value). The 108 genes showed intermediate relative expression levels in the low viral load group and low expression in the SARS-CoV-2 negative control group. Of the 429 genes that correlated with extremely high viral load at p<0.01, 367 genes were highly enriched for functional gene categories, as defined by GO annotation terms. Enriched GO terms (Fig 3B, p< = 3E-5, one-sided Fisher's exact test) included 'immune system process', 'response to virus', 'type I interferon signaling pathway', 'cytokine-mediated signaling pathway', 'response to stress', 'regulation of viral life cycle', 'immune response', 'response to cytokine', 'innate immune response', 'response to interferon-gamma', 'regulation of I-kappaB kinase/NF-kappaB signaling', 'JAK-STAT cascade', 'protein ubiquitination', 'regulation of cell death', 'T cell activation', 'vesicle-mediated transport', and 'complement activation'. Of the 17 functional gene categories, there were five gene categories—'response to virus', 'type 1 interferon signaling', 'regulation of viral life cycle', 'response to interferon-gamma', and 'JAK-STAT cascade'–where approximately 20% or higher of the genes were over expressed for that pathway. Overall, the above gene categories were highly indicative as representing a host immune response to an acute viral infection. Some of the genes that were upregulated were *EIF2AK2* (eukaryotic translation initiation factor 2 alpha kinase 2), and *ZC3HAV1* (zinc finger CCCH-type containing, antiviral 1), which have anti-viral activity. Other genes like *IFIT2* and *IFIT3* (interferon induced protein with tetratricopeptide repeats) aid in apoptosis. Chemokine

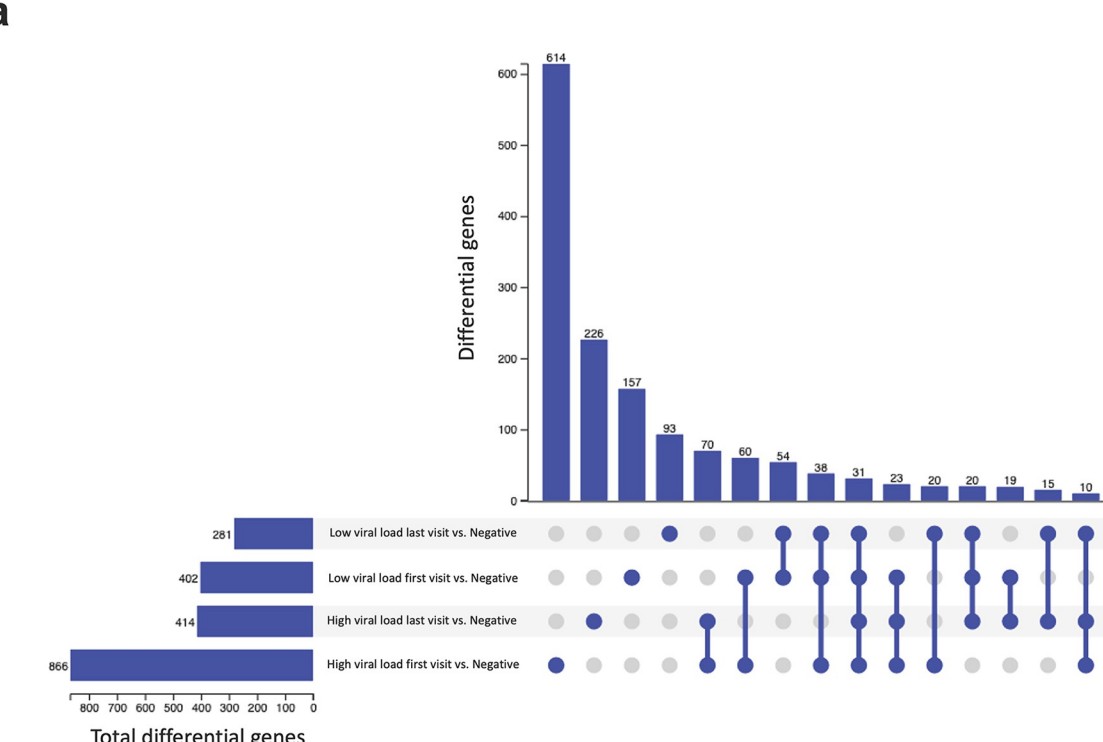

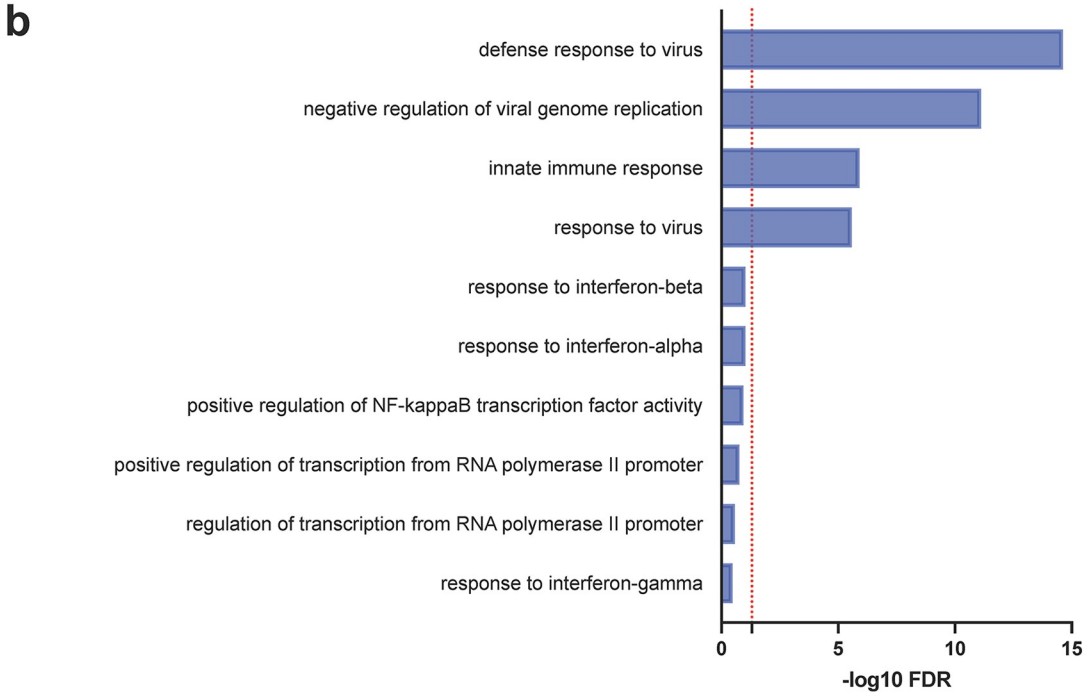

**Fig 2. UpSet plot summarising key differentially expressed gene trends and Gene ontology analysis of differentially expressed genes present only in extremely high viral load group. (a)** The plot depicts common and unique genes shared between extremely high viral load group and low viral load group as compared to SARS-CoV-2 negative group groups. Set size indicates number of differentially expressed genes in each comparison. Intersection size is the number of statistically significant (FDR < 0.05) differentially expressed genes in designated sets or groups. Connected circles at the bottom of the plot indicate an intersection of

differentially expressed genes between groups. **(b)**The top10 enriched GO terms for biological processes altered in extremely high viral load group in shown. Significantly enriched GO terms with a minimum three enriched genes were ranked by significance. The x-axis denoting the negative log fold change of significance. Dotted red line depicts the significance threshold of FDR <0.05.

genes like *CXCL9* and *CXCL10* that are involved in T-cell trafficking were also highly expressed. In contrast, the 62 genes decreased with high viral load at p<0.01 were not highly enriched for GO terms. Some of the genes that were downregulated included *OR4A16* and *OR10X1*, involved in olfactory responses; *SALL3* and *MAGB6*, which aid in downregulation of transcription; and *TUBA3E*, *MLN*, and *ISTN1*, affecting tubulin functions.

## Comparison of the top over expressed genes in respiratory samples from the extremely high and low viral load groups to other published data sets

Genes correlated with SARS-CoV-2 viral load over time were similarly differentially expressed across independent datasets of SARS-CoV-2 infected lung and upper airway cells (Fig 4). We examined differential expression patterns for the top 112 genes, at p<0.001 significance level, correlated with SARS-CoV-2 viral load across our serial sampling cohort (by Pearson's) in two independent RNA-seq datasets of SARS-CoV-2 infection: one of lung cancer cell lines A549 and Calu-3 infected with SARS-CoV-2 for 24 hours from Blanco-Melo et al [28] and one of nasopharyngeal/oropharyngeal samples in 238 patients with COVID-19, other viral, or non-viral acute respiratory illnesses from Mick et al [29] (these two datasets representing some of the earliest studies of SARS-CoV-2 host responses). As a group, the genes that positively corre-lated with SARS-CoV-2 viral load were increased in SARS-CoV-2-infected Calu-3 cells and were high in samples of human subjects infected with SARS-CoV-2 or other viruses (Fig 4A). For the Mick et al. dataset, SARS-CoV-2 viral load data was available. Of the 112 genes corre-lated with viral load in our dataset, 105 were in common with the Mick et al. dataset, and 99 (94%) of these genes were positively correlated (Pearson's p<0.05) with viral load across the 94 SARS-CoV-2 infected patients. In contrast to Calu-3, A549 infected cells did not show as strong a correspondence to our 112-gene signature pattern. Taking the top genes that corre-lated positively with SARS-CoV-2 viral load across the Mick et al. patient samples (p<0.01, Pearson's correlation) and the top genes over-expressed in SARS-CoV-2-infected Calu-3 cells (p<0.01, t-test), these significantly overlapped with the genes that positively correlated (p<0.01, Pearson's) with SARS-CoV-2 viral load across our serially collected MT swab samples with a highly statistically significant overlap among the respective dataset results (Fig 4B). The 136 genes overlapping among all three datasets involved cytokines and inflammatory response pathways. In contrast, there was limited overlap among the datasets involving genes under-expressed with SARS-CoV-2 infection between our data set to either Mick *et al.* or Calu-3 cells (Fig 4C).

## Comparison of our respiratory sample gene sets to the transcriptional response of the human nose organoid infected with SARS-CoV-2

As another means to identify host transcriptional responses to SARS-CoV-2 infection, we gen-erated RNA-seq data on the human nose organoid model HNO [30]. We sampled HNO cells infected with SARS-CoV-2 and mock control cells at 6hrs, 72hrs, and 6 days post-infection, and we profiled these samples for gene expression. In the HNO204 RNA-seq dataset, 1760 genes were statistically significant at p<0.05 significance level and 341 genes, at p<0.01, exceeding chance expected. The top 867 genes over-expressed in HNO with SARS-CoV-2 infection (p<0.05, t-test) showed significant overlapping patterns with the above-mentioned

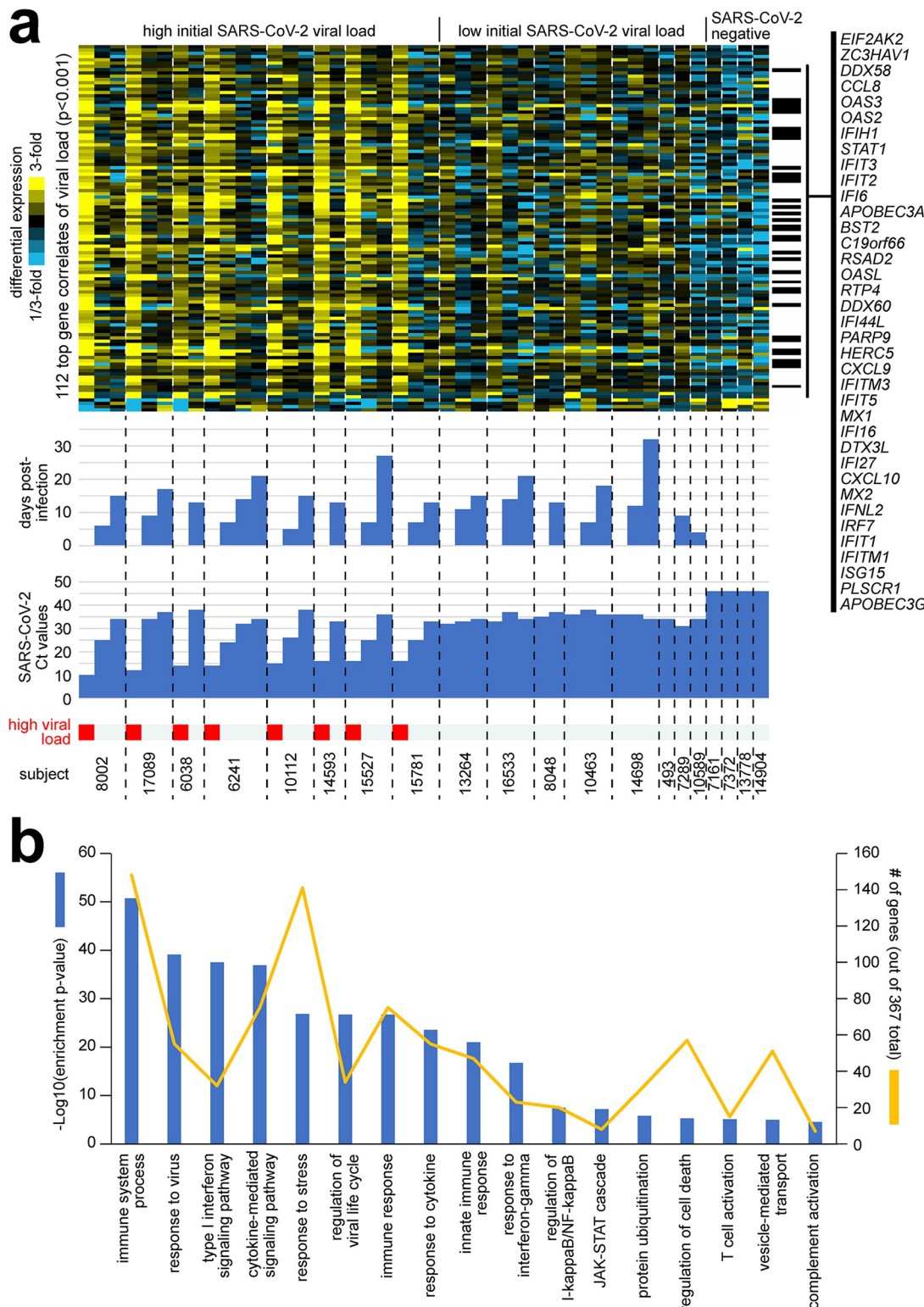

**Fig 3. Differential expression patterns and functional gene groups associated with SARS-CoV-2 viral load across serially collected samples. (a)** Across 44 MT swab samples representing 20 subjects, differential gene expression patterns for the set of 112 genes significantly correlated with SARS-CoV-2 viral load (i.e., inversely correlated with Ct value) at p<0.001 (Pearson's) are represented. Heat map contrast (bright yellow/blue) is 3-fold change from the average of the samples from the low viral load group. Genes listed off to the right have GO annotation 'response to virus'. Extremely high viral load, Ct<20. Visualization using

heat maps was performed using JavaTreeview (version 1.1.6r4) [27] **(b)** Selected significantly enriched GO terms [26] within the genes over-expressed with SARS-CoV-2 viral load (p<0.01, Pearson's). For each GO term, enrichment p-values and numbers of genes in the SARS-CoV-2-associated gene set are indicated. Enrichment p-values by one-sided Fisher's exact test.

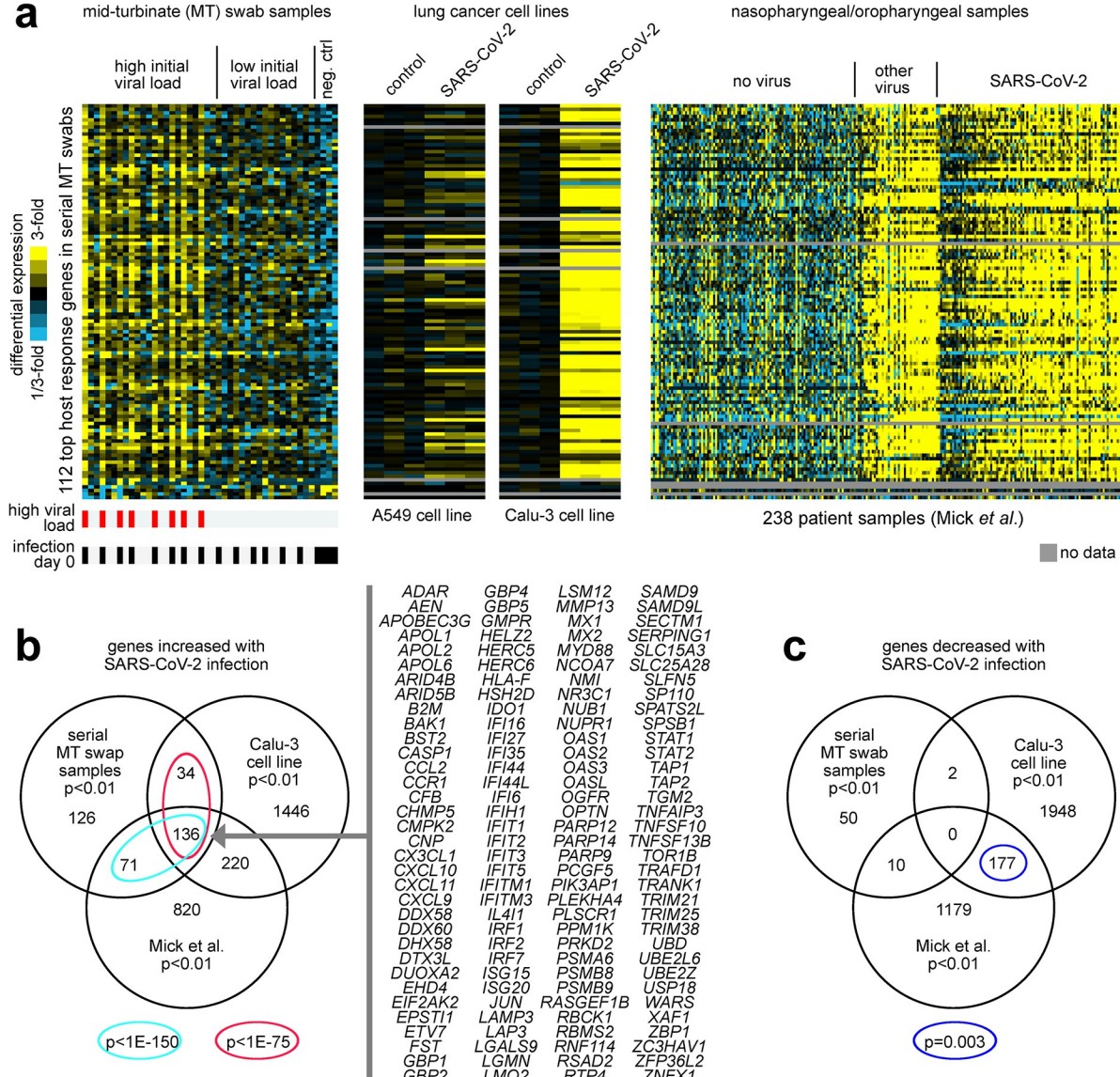

**Fig 4. Genes correlated with SARS-CoV-2 viral load over time are similarly expressed in independent datasets of SARS-CoV-2 infected lung and upper airway cells. (a)** Differential expression patterns for the 112 genes correlated with SARS-CoV-2 viral load across our serial sampling cohort (p<0.001, from Fig 3A) were examined in two independent RNA-seq datasets of SARS-CoV-2 infection: one of lung cancer cell lines (A549 and Calu-3) infected with SARS-CoV-2 at multiplicity-of-Infection (MOI) of 2 for 24 hours (in biological triplicate)[28], and one of nasopharyngeal/oropharyngeal samples in 238 patients with COVID-19, other viral, or non-viral acute respiratory illnesses [29]. Gene order is the same across all datasets. Heat map contrast (bright yellow/blue) is 3-fold change from the corresponding comparison group (serial sampling dataset, average of the samples from the low viral load group; lung cancer cell line dataset, average of corresponding mock control group; Mick et al. dataset, average of "no virus" samples). Visualization using heat maps was performed using JavaTreeview (version 1.1.6r4) [27] **(b)** Venn diagram representing the gene set overlaps among the genes increased with SARS-CoV-2 infection in each of the three RNA-seq datasets from part a (with Calu-3 lung cancer cell line being considered here over A549). A p-value cutoff of p<0.01 was used to define top genes for each dataset (serial MT swab and Mick et al. nasopharyngeal/oropharyngeal datasets, Pearson's correlation with viral load; Calu-3 dataset, t-test). Gene set enrichment p-values by one-sided Fisher's exact test. Genes overlapping between all three datasets are listed. **(c)** Similar to part b, but for genes decreased with SARS-CoV-2 infection.

independent RNA-seq datasets of SARS-CoV-2 infection (Fig 5A). Only a small, albeit statistically significant, fraction of the HNO204 over-expressed genes overlapped with the top 367 genes that correlated positively with SARS-CoV-2 viral load in our serial MT swab dataset (Fig 5B). Of the 867 overexpressed genes in SARS-CoV-2 infected HNO, 35 overlapped with the 367 over expressed genes in the respiratory samples of extremely high and low viral load groups (p = 1E-5, one-sided Fisher's exact test). At the same time, a substantial fraction of the 867 HNO genes overlapped with the genes high with SARS-CoV-2 infection in both A549 and Calu-3 lung cancer cell lines (Fig 5A and 5B), with 178 Calu-3 genes overlapping (p<1E-20, one-sided Fisher's exact test). In contrast, little overlap was observed between the genes under-expressed with SARS-CoV-2 infection in HNO and genes similarly under-expressed with SARS-CoV-2 in the other datasets (Fig 5C).

The 867 genes over-expressed in HNO at p<0.05 were significantly enriched for functional GO gene categories. Enriched GO terms (Fig 5D, p< = 0.0001, one-sided Fisher's exact test) included 'vesicle', 'extracellular vesicle', 'intracellular vesicles', 'MAP kinase phosphatase activity', 'regulation of locomotion', 'peptidase activator activity', 'endosome membrane', 'regulation of smooth muscle cell proliferation', 'regulation of cell motility', 'proteasomal protein catabolic process', 'negative regulation of signaling', 'programmed cell death', proteolysis involved in cellular protein catabolic process', and 'inactivation of MAPK pathway'. Overall, the over expressed genes are representative of the regulation of extracellular signaling from virus infection on a wide range of cellular responses and function. The above findings of the HNO transcriptional response to SARS-CoV-2 in relation to transcriptional responses observed in other models and patient samples would suggest the existence of distinct host responses to SARS-CoV-2 depending on cellular context, such as we previously observed between A549 and Calu-3 lung cancer cell lines. The host response observed in HNO is reflective of a complex epithelial cell population responding to a SARS-CoV-2 infection. On the other hand, the host response genes detected in the upper respiratory tract secretion of our prospective longitudinal cohort and those of Mick *et al.* patient samples are a composite of the epithelial and cellular immune responses to the viral infection.

## Discussion

The primary site for SARS-CoV-2 replication is thought to be the ciliated cells in the nasopharynx or nasal olfactory mucosa. The viral replication initiates a signaling cascade to promote the production of interferons and chemokines by epithelial cells and thereby promote immune cell activation to control the virus. SARS-CoV-2 infection causes upregulation of cytokines including IL-2, IL-6, IL-10, IL-12 and MCP-1 detected in tissues and serum, as well as infiltration of infected tissues by inflammatory cells such as macrophages [31]. In the present study, RNA seq analysis of MT swabs from SARS-CoV-2 infected individuals identified robust induction of interferon inducible, cytokine, stress response, and immune-related genes. A variety of genes such as *OAS2*, *PARP9*, *OASL*, *IFIT2*, *IFI3*, *CCL8*, *CXCL10*, etc., were highly upregulated and correlated with high viral load, suggesting that innate immune response genes were activated in a viral load dose response manner to control the viral infection. These results are very consistent with recent studies from upper respiratory tract samples, which reported upregulation of anti-viral factors and interferon response pathways [22,24,32].

In our study samples, the numbers of genes that were upregulated were much higher compared to down regulated genes (367 vs 62). Some of genes that were downregulated included those which operate olfactory functions (*OR4A16* and *OR10X1*), downregulation of transcription (*SALL3* and *MAGB6*), and tubulin functions (*TUBA3E* and *MLN*, and *ISTN1*). Previous studies have reported larger numbers of down regulated host response genes especially

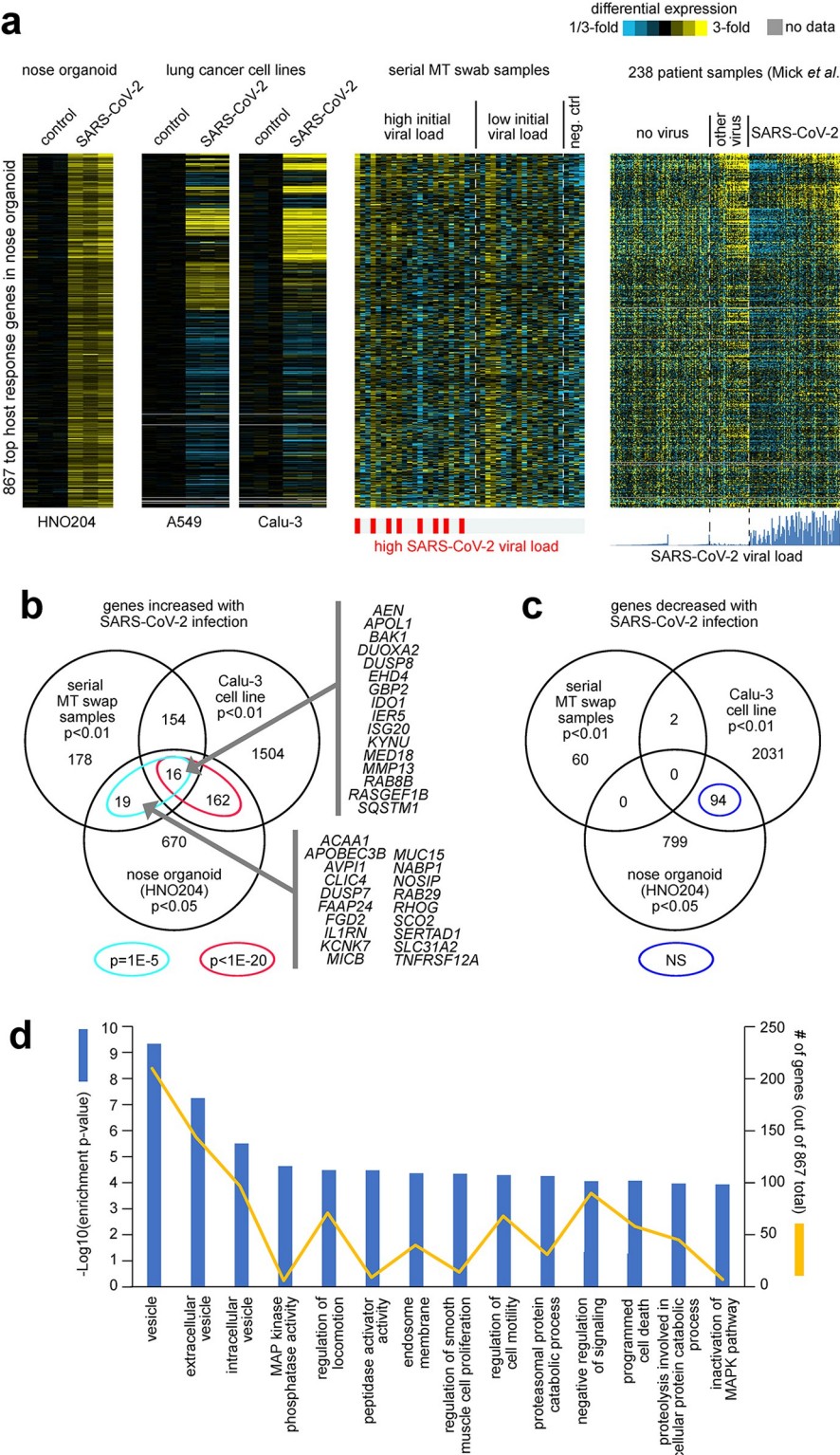

**Fig 5. Differential expression patterns and functional gene groups associated with SARS-CoV-2 infection of nose organoids. (a)** HNO204 human nose organoids were infected with SARS-CoV-2 at an MOI of 0.01, and samples at 6hrs, 72hrs, and 6 days post infection were profiled for gene expression. Differential expression patterns for the top 867 genes over-expressed in HNO204 with SARS-CoV-2 infection (p<0.05, t-test) are represented here. Next to the HNO204 dataset are the corresponding patterns for independent RNA-seq datasets of SARS-CoV-2 infection: lung

cancer cell lines (A549 and Calu-3)[25], our serially collected MT swab samples from patients, and nasopharyngeal/oropharyngeal samples from Mick et al [29]. Gene order is the same across all datasets. Heat map contrast (bright yellow/blue) is 3-fold change from the corresponding comparison group. Regarding the Mick *et al.* dataset, the ordering of the samples is the same as that for Fig 4A, and the SARS-CoV-2 viral load plot represents log2 RPM, with values ranging from 0 to 19.6. Visualization using heat maps was performed using JavaTreeview (version 1.1.6r4) [27] **(b)** Venn diagram representing the gene set overlaps among the genes increased with SARS-CoV-2 infection in each of the following RNA-seq datasets: HNO204, serial MT swab, and Calu-3 lung cancer cell line. Gene set enrichment p-values by one-sided Fisher's exact test. Genes overlapping between HNO204 and serial MT swab datasets are listed. **(c)** Similar to part b, but for genes decreased with SARS-CoV-2 infection. **(d)** Selected significantly enriched GO terms [41] within the genes over-expressed with SARS-CoV-2 infection in HNO204 (p<0.05, t-test). For each GO term, enrichment p-values and numbers of genes in the SARS-CoV-2-associated gene set are indicated. Enrichment p-values by one-sided Fisher's exact test.

involving olfactory receptor pathway, neutrophil degranulation, and vesicle formation—indicating the role of these genes in loss of olfactory function in SARS-CoV-2 infections as well as the viral control of host-cell machinery [20,22,24]. One other study also showed very low number of downregulated genes with SARS-CoV-2 infection [33], one reason for the low number of downregulated genes observed in our longitudinal study could conceivably relate to the timing of sample collection as compared to other studies.

Remarkably, the highest number of significant expressed genes were driven by the extremely high viral load group at Visit 1 (first visit). Also, all the genes that were upregulated with the low viral load group at Visit 1 completely overlapped with the extremely high viral load group at Visit 1 except for one gene, -*CNN2*, which plays a role in cell adhesion and muscle contraction. The predominant sets of genes involved in defense response to virus, type I interferon signaling pathway, cytokine-mediated signaling pathway—such as *CXCL10*, *TGFB*, *IFIT2*, *IFIT3*, *OAS1*, and *IRF1*—were not found significantly upregulated in the low viral load group. Consonant with this, Rouchka *et. al.* also observed that subjects with high viral loads had robust interferon and cellular anti-viral response and even exhibited strong inverse correlation with disease severity [6]. We previously noted that some SARS-CoV-2 infected adults with low viral load experienced prolonged viral shedding and low fluctuation in viral load over time [14]. Absence or low expression of the anti-viral response in the low viral load group strengthens our observation of prolonged shedding in adults with a low viral load early in infection.

In our longitudinal study, the up-regulated host response genes that correlated with SARS-CoV-2 viral load over time in the respiratory secretion collected by the MT swabs were similarly differentially expressed across independent data sets of SARS-CoV-2 infected lung and upper airway cells [28]. About 170 of the differentially expressed genes observed in our study overlapped with SARS-COV-2 infected Calu-3 lung adenocarcinoma cell line but not with A549 cells. The observed difference across the cell lines could possibly be attributed to A549 cells not supporting robust replication of SARS-CoV-2 due to the low expression of ACE-2 [34]. Similarly, 207 up-regulated genes from our longitudinal study overlapped with nasopharyngeal swabs from SARS-CoV-2 infected patients (3). Genes involved in cytokines and inflammatory response pathways were the ones that overlapped the most, demonstrating that anti-viral innate immune responses are common with SARS-CoV-2 infections. In addition, the up-regulation of differentially expressed genes related to an inflammatory response in COVID-19 patients can result in the induction of interleukin-6 (IL-6), CXCL10 (IP-10), and TNF-α with hyperactivation of Th1/Th17 responses that results the recruitment and activation of pro-inflammatory neutrophils and macrophages into the airways [35]. This has been proposed as the prime reason for failure to resolve inflammation in severely symptomatic patients [35,36].

To better understand the contribution of epithelial cellular responses to SARS-CoV-2, we compared differentially expressed genes in the respiratory secretion of adults infected with SARS-CoV-2 to those that were expressed in HNO infected with SARS-CoV-2. A small, albeit statistically significant, fraction (35 of 867) of the HNO up-regulated genes overlapped with the 367 differentially expressed up-regulated genes detected from the SARS-CoV-2 cases from our longitudinal cohorts. These included functional genes involved with intrinsic antiviral immunity and interferon signaling representing the epithelial cellular responses to SARS-CoV-2 infection. A greater number of up-regulated genes overlap between our longitudinal cohorts [170 (46.3%) of 367 genes] and the SARS-CoV-2 infected Calu-3 cell line [170 (9.3%) of 1836 genes] compared to our SARS-CoV-2 infected HNO204 line [35 (4.0%) of 867 genes]. HNO204 is a complex pseudostratified epithelium composed of at least 9 different cell types including ciliated, goblet, secretory and basal cells [30,37]. However, it does lack the immune cells that play an important role in the regulation of viruses. The fewer overlap between HNO data set and MT swab could in part be due to lack of immune cells in the HNO and presence of these in nasal samples. The HNO data represents 6 hours, 3 days and 6 days post SARS-CoV-2 infection. However, all the time points were combined to compare to other data sets, which could have dampened fold change. SARS-CoV-2 infects a subset of the apical ciliated cells and generates significantly lower cytokine response compared to RSV [30]. In contrast, the Calu-3 cell line, was generated from a bronchial adenocarcinoma, a submucosal gland cell line of a single cell type [38] that is highly susceptible to SARS-CoV-2 infection thereby almost all the cell population is responding to the infection and could account for higher overlap of the genes with MT swabs.

Previous studies have demonstrated high expression of ACE2 in SARS-CoV-2 infected nasopharyngeal samples and these were greatly elevated in high viral load subjects, suggesting that higher replication occurs with increased receptor expression [22]. In our cohort we did not observe a statistically significant increase in *ACE2* expression in both extremely high and low viral load groups. However, the expression of *ACE2* was elevated in our HNO infected with SARS-CoV-2 but not *TMPRSS2*, which has increased expression in nasal airway epithelial brushings [39].

In summary, our longitudinal study investigated gene expression patterns in SARS-CoV-2 infected individuals with an extremely high viral load displayed strong immune responses that decreased over time, and eventually became comparable to those with low viral loads. We detected hundreds of up-regulated genes that were highly correlated to the SARS-CoV-2 viral load. Enriched cellular pathways involved in the innate immune response, antiviral interferon responses were observed in other cohorts of SARS-CoV-2 infected adults. A limited but highly significant up-regulated gene response overlapped with our human nose organoid line, a complex pseudostratified ciliated epithelium, suggesting that the gene expression profile detected in SARS-CoV-2 infected adults is generated from both the epithelial and cellular immune responses. In conclusion, high SARS-CoV-2 viral loads primarily elicit a heightened host immune response for the control of viral replication and clearance.

Some limitations of our study include the small sample size and the generalizability of our findings to larger and more diverse populations. A number of factors can significantly impact the host response to infection, including age, sex, comorbidities, and genetics. Individuals with pre-existing health conditions may have altered gene expression profiles that affect their response to infection, leading to increased morbidity and mortality. In addition, enrolling a control group with no infection or a different viral infection may unravel pathways unique to SARS-COV-2 and provide avenues for deeper exploration of how observed gene expression changes affect immune responses and disease outcomes. As future work, gene expression profiling of larger and more diverse patient cohorts with an extended follow-up period would

enhance the generalizability of gene expression signatures of our present study. It is worth noting however, that a significant portion of the genes in our transcription signature, related to responses to SARS-CoV-2 infection, were found to be expressed in an independent patient group, as well as in model systems such as lung cancer cell lines and nose organoids. Understanding the interplay between viral dynamics and host responses is essential for resolving the pathogenesis of COVID-19, leading to better management and prevention strategies.

## Materials and methods

### Ethical approval

RT-PCR testing was performed as a service to BCM, the collection of metadata was performed under an Institutional Review Board of Baylor College of Medicine approved protocol (H-47423) with waiver of informed consent. All methods were performed in accordance with the relevant guidelines and regulations.

### Study cohort

Ten extremely high, viral load SARS-CoV-2 positive cases were matched to 10 low viral load SARS-CoV-2 positive adults, and 10 stable adults (SARS-CoV-2 negative controls) who were cleared for having an out-patient surgical or aerosol generating procedure. The cases and controls were selected from our population of 17,644 adults (24,822 samples) evaluated in the outpatient clinics at Baylor College of Medicine (BCM) and their affiliate institutions from March 18, 2020, through January 16, 2021, as previously described[40].Three distinct adult populations were tested: 1) symptomatic employees utilizing occupational health services, 2) patients evaluated at medical and surgical clinics, and 3) patients who required clearance for an outpatient surgical or aerosol generating procedure. Serial samples were obtained from individuals who came back to be tested for evidence that the virus was cleared or were enrolled as substudy to determine the viral shedding kinetics. Testing for SARS-CoV-2 was performed in our Clinical Laboratory Improvement Amendments (CLIA) Certified Respiratory Virus Diagnostic Laboratory (ID#: 45D0919666).

The extremely high viral load cases consisted of adults with an extremely high viral load (Ct <16) for the N1 target on their first mid-turbinate (MT) sample and had at least two subsequent positive MT samples [14]. Of the 104 individuals with an extremely high viral load in their first test, 30 individuals met the criteria for multiple positive samples over the ensuing 4 weeks. Adults from two other groups were matched to each extremely high viral load case: a low viral load (Ct 31-<40) SARS-CoV-2 positive adult (SARS-CoV-2 low viral load) and an otherwise stable control who tested negative for SARS-CoV-2 (SARS-CoV-2 negative control), was asymptomatic for respiratory infection and was cleared for an out-patient surgical or aerosol generating procedure. Of the 453 individuals with a low viral load in their first test, 126 individuals met the criteria for multiple positive samples over the ensuing 4 weeks. The extremely high viral load cases were matched to the other two groups by gender, week of first test (+ 1 week), age (+ 1 year) and zip code (5 digits). If a match could not be found the range of the factors were expanded to + 3 weeks of first test, + 10 years and 3 digits for the zip code. The ten extremely high viral load cases were randomly selected from our pool of 30 individuals with an extremely high viral load with multiple positive MT samples. The best matched SARS-CoV-2 low viral load case and negative control were then selected for each extremely high viral load case. For each high viral load case two to three subsequent SARS-CoV-2 RT-PCR positive mid-turbinate (MT) swab samples were collected over a 4-week period. Each SARS-CoV-2 low viral load case had similarly spaced SARS-CoV-2 positive MT swab samples matched to its respective extremely high viral load case. On the other hand, the SARS-CoV-2 negative control

only had one MT-swab sample collected with no longitudinal follow-up and was used to establish the transcriptomic baseline in the respiratory epithelium during the time the extremely high and low viral load matched cases were identified.

## RNA sequencing of serially collected specimens

RNA sequencing data is available on eight cases (extremely high viral load) with 23 samples. Six of the 8 extremely high viral load cases had gene expression data for Visits 1, 2, and 3, and two others for Visit 1 and 3. On the other hand, eight low viral load cases had 17 samples with gene expression data. Only two of the low viral load cases had gene expression data for Visit 1, 2, and 3. Another two low viral load cases had gene expression data at Visit 1, Visit 2 or 3, and Visit 4. The remaining four low viral load cases had gene expression data at Visit 1 only (n = 1), Visit 2 only (n = 2) or Visit 1 and 2 (n = 1). Only 4 of the 10 SARS-CoV-2 negative control adults had gene expression data. All together 44 MT swab samples were sequenced to observe gene expression changes.

## SARS-CoV-2 RT- PCR

Viral RNA extraction and RT-PCR testing was performed as previously described (Avadhanula et al., 2021). In brief, viral RNA was extracted using the Qiagen Viral RNA Mini Kit (QIAGEN Sciences, Maryland, USA) with an automated extraction platform QIAcube (QIAGEN, Hilden, Germany). The extracted RNA samples were tested by CDC 2019-novel coronavirus (2019-ncoV) Real-Time RT-PCR Diagnostic panel [CDC 2019-Novel Coronavirus (2019-nCoV) Real-Time RT-PCR Diagnostic Panel for Emergency Use Only Instructions for Use]. RT-PCR reaction was set up using TaqPath™ 1-Step RT-qPCR Master Mix, CG (Applied Biosystems, CA) and run on 7500 Fast Dx Real-Time PCR Instrument with SDS 1.4 software. Respiratory samples with cycle threshold (Ct) values <40 for both N1 and N2 primers were considered RT-PCR positive for SARS-CoV-2.

## Human nose organoid model

The differentiated human nose organoid derived air liquid interface (HNO-ALI) cells were apically infected with SARS-CoV-2 [Isolate USA-WA1/2020, obtained from Biodefense and Emerging Infectious resources (BEI)] at a multiplicity of infection of 0.01 or mock infected with airway organoid differentiation media, as previously described[30]. At the respective time points, the apical side of the transwells was washed twice and the cells were lysed using lysis buffer of RNeasy mini kit and RNA extracted.

## RNA extraction, library preparation and sequencing

Samples were extracted using the Qiagen RNeasy mini kit (#74104 rev. 10/19) following the manufacturer's protocol for samples <5e6 cells. Samples were eluted in 50ul RNase-free water. RNA quality and quantity were estimated using Agilent Bioanalyzer OR Caliper GX. To monitor sample and process consistency, 1 μl of the 1:50 diluted synthetic RNA designed by External RNA Controls Consortium (ERCC) (4456740, ThermoFisher) was added. Whole transcriptome sequencing (total RNAseq) data was generated using the Illumina TruSeq Stranded Total RNA with Ribo-Zero Globin kit (20020612, Illumina Inc.) cDNA was prepared following rRNA and Globin mRNA depletion, and paired-end libraries were prepared on Beckman BioMek FXp liquid handlers. For this, cDNA was A-tailed followed by ligation of the TruSeq UD Indexes (Cat # 20022370) and amplified for 15 PCR cycles following manufacturer's recommendation. AMPure XP beads (A63882, Beckman Coulter) were used for library

purification. Libraries were quantified using a Fragment Analyzer (Agilent Technologies, Inc) electrophoresis system and pooled in equimolar ratios. This pool was quantified using qPCR to determine loading concentration for sequencing. Sequencing was performed on the Nova-Seq 6000 instrument using the S4 reagent kit (300 cycles) to generate 2x150bp paired end reads.

### Primary analysis for total RNASeq

The RNA-Seq analysis pipeline cleans and processes raw RNA sequencing data (FASTQs), providing robust QC metrics and has the flexibility to map the reads to GRCh38 reference genome (after excluding the alternate contigs). The latest versions of software for sequence alignment (STAR v2.7.3a), for marking of duplicate reads (Picard v2.22.5) and for conversion of BAM files to FASTQ files (Samtools v.1.9) are part of this pipeline. In addition to these components, the pipeline uses RSEM (v.1.3.3) for measuring gene expression and RNA-SeQC (v.1.1.9), Qualimap2 (v2.2.1) and ERCCQC (v.1.0) to generate quality control metrics on the RNA-Seq data. The pipeline also produces the raw gene features counts by using feature-Counts (v2.0.1).

### Gene expression analysis

For our serial MT swab dataset, where RNA-seq data were generated in different batches involving time and differences in extraction and processing methods, Combat algorithm [41] was used to correct for any observed batch effects. Fragments Per Kilobase Million (FPKM) values were quantile normalized [42] and log2-transformed. The differential expression analyses to define the host transcriptional response focus on 20000 genes for which an Entrez identifier could be associated with the transcript feature. To identify expression patterns associated with the host transcriptional response to SARS-CoV-2 infection in the serial MT swab dataset, Log2 expression values correlated with SARS-CoV-2 Ct values across all 44 samples in the RNA-seq dataset (S1 File). Additional two-group comparisons were carried using t-test on log2 expression values. For the HNO204 nose organoid dataset, Log2 expression values were compared between SARS-CoV-2 and Mock control by t-test, combining time points of 6hrs, 72hrs, and 6 days for each group.

### Analysis of external transcriptome datasets

To define transcriptional signatures of the host cell response to SARS-CoV-2 infection in lung cancer cells, we referred to the GSE147507 RNA-seq dataset [28]. In this dataset, A549 and Calu-3 were mock-treated or infected with SARS-CoV-2 and then profiled for gene expression. We used data from the SARS-CoV-2 profiling experiments involving multiplicity-of-Infection (MOI) of 2. We converted raw gene-level sequencing read counts to reads per million Mapped (RPM) values and then log2-transformed them [43]. For the Mick et al. RNA-seq dataset of nasopharyngeal/oropharyngeal samples in 238 patients with COVID-19, other viral, or non-viral acute respiratory illnesses [29], RPM values were quantile normalized before the analysis.

### Statistical analysis

All p-values were two-sided unless otherwise specified. We performed all tests using log2-transformed gene expression values. False Discovery Rates (FDRs) due to multiple testing of genes were estimated using the method of Storey and Tibshirini [25]. Even in instances of nominally significant genes only moderately exceeding chance expectations by FDR, the

nominally significant genes were found in downstream enrichment analyses (involving functional gene sets and results of external SARS-CoV-2-related RNA-seq datasets) to contain biological information. We evaluated enrichment of GO annotation terms [26] within sets of differentially expressed genes using SigTerms software [44] and one-sided Fisher's exact tests. Visualization using heat maps was performed using JavaTreeview (version 1.1.6r4) [27] and matrix2png (version 1.2.1) [45] Gene ontology (GO) analysis of DEGs used in the UpSet plot (Fig 2) was performed using the web-based Database for Annotation, Visualization, and Integrated Discovery (DAVID; version—v2023q1) [46,47].

## Supporting information

**S1 Table. Demographic and visit characteristics by initial matched groups.**
(DOCX)

**S1 File.**
(XLSX)

## Author Contributions

**Conceptualization:** Vasanthi Avadhanula, Chad J. Creighton, Pedro A. Piedra.

**Data curation:** Vasanthi Avadhanula, Chad J. Creighton, Sara Joan Javornik Cregeen.

**Formal analysis:** Chad J. Creighton, Divya Nagaraj, Yiqun Zhang.

**Funding acquisition:** Vasanthi Avadhanula, Chad J. Creighton, Richard A. Gibbs, Joseph F. Petrosino.

**Investigation:** Vasanthi Avadhanula, Chad J. Creighton, Laura Ferlic-Stark, Pedro A. Piedra.

**Methodology:** Vasanthi Avadhanula, Chad J. Creighton, Laura Ferlic-Stark, Divya Nagaraj, Richard Sucgang, Erin G. Nicholson, Anubama Rajan, Vipin Kumar Menon, Harshavardhan Doddapaneni, Donna Marie Muzny, Ginger A. Metcalf, Sara Joan Javornik Cregeen, Kristi Louise Hoffman.

**Project administration:** Richard Sucgang, Vipin Kumar Menon, Harshavardhan Doddapaneni, Donna Marie Muzny, Ginger A. Metcalf, Sara Joan Javornik Cregeen, Kristi Louise Hoffman.

**Resources:** Harshavardhan Doddapaneni.

**Supervision:** Vasanthi Avadhanula, Chad J. Creighton, Richard A. Gibbs, Joseph F. Petrosino.

**Visualization:** Chad J. Creighton, Laura Ferlic-Stark, Divya Nagaraj, Yiqun Zhang, Pedro A. Piedra.

**Writing – original draft:** Vasanthi Avadhanula, Chad J. Creighton, Pedro A. Piedra.

**Writing – review & editing:** Vasanthi Avadhanula, Chad J. Creighton, Laura Ferlic-Stark, Divya Nagaraj, Erin G. Nicholson, Anubama Rajan, Vipin Kumar Menon, Harshavardhan Doddapaneni, Donna Marie Muzny, Ginger A. Metcalf, Sara Joan Javornik Cregeen, Kristi Louise Hoffman, Richard A. Gibbs, Joseph F. Petrosino, Pedro A. Piedra.

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
