## [Decision Letter · Decision Letter 0]

27 Sep 2024

PONE-D-24-25457Longitudinal host transcriptional responses to SARS-CoV-2 infection in adults with extremely high viral loadPLOS ONE

Dear Dr. Avadhanula,

Thank you for submitting your manuscript to PLOS ONE. After careful consideration, we feel that it has merit but does not fully meet PLOS ONE’s publication criteria as it currently stands. Therefore, we invite you to submit a revised version of the manuscript that addresses the points raised during the review process.

**ACADEMIC EDITOR: Please address reviewers concerns as soon as possible.**

We look forward to receiving your revised manuscript.

Kind regards,

Ravikanth Nanduri, Ph. D.

Academic Editor

PLOS ONE

Journal Requirements:

“This work was supported by NIH grants CA125123 (CJC), U19AI144297 (VA, CJC, RG, JP, PAP)”

“U19AI144297

CA125123”

Reviewers' comments:

Reviewer's Responses to Questions

**Comments to the Author**

1. Is the manuscript technically sound, and do the data support the conclusions?

Reviewer #1: Yes

Reviewer #2: Partly

2. Has the statistical analysis been performed appropriately and rigorously? 

Reviewer #1: Yes

Reviewer #2: Yes

3. Have the authors made all data underlying the findings in their manuscript fully available?

Reviewer #1: Yes

Reviewer #2: Yes

4. Is the manuscript presented in an intelligible fashion and written in standard English?

Reviewer #1: Yes

Reviewer #2: Yes

5. Review Comments to the Author

Reviewer #1: This study by Avadhanula et al. assessed the differential pattern in level of gene expression in different group of SARS-COV-2 patients vs healthy people. Using strong analysis methods, the authors could show the difference in gene expression level in different groups, which correlated with the viral load and the severity of the state. More interestingly, the authors showed that in the first days of the infection, robust protective immune response-associated gene are up regulated and correlated with the severity of the infection and the viral load. With the clearance of the infection over the time, the viral load decreases and correlate with a reduction of immune response-related genes, and the gene expression pattern changes. This study not only show the genomic pattern over the infection, but also provides an insight that could help in the prognostic of the severity of the infection.

The specimens in the study were well recruiting, the methods clearly described and justified, and the results were presented in a clear and scientific way. The discussion of this manuscript supports the generated results. This study would be of a great interest, specially for clinical setup.

Reviewer #2: This study provides valuable insights into the dynamic relationship between SARS-CoV-2 viral load and host gene expression of SARS-COV2 illness. Examining patients with varying viral loads has identified distinct transcriptional responses that correlate with the severity of infection. Comparing gene expression across high viral load, low viral load, and asymptomatic cases is supreme, as this approach is instrumental in understanding the complexities of COVID-19 gene expression Patterns; the distinct gene expression profiles in high viral load cases that have been identified hold the potential to indicate pathways contributing to increased viral replication or immune evasion. This promising finding could pave the way for identifying intervention targets, offering a ray of hope for more effective COVID-19 treatments. This response diminished over time as viral loads decreased, suggesting a temporal relationship between viral presence and host immune activation.

Differential expression across contexts; the genes correlated with viral load showed similar expression patterns in independent datasets, including lung and upper airway cells. This consistency underscores the potential universality of the host response to SARS-CoV-2 across different tissue types, providing a solid foundation for further research and interventions. Using the nose organoid model, the study team replicated many host responses observed in patient samples. This model offered additional evidence that the context of the cellular environment significantly influences the nature of the immune response to infection, enlightening us on the crucial role of the cellular environment in the immune response to SARS-CoV-2. The findings indicate that both epithelial and immune cell responses are involved in the host reaction to SARS-CoV-2, highlighting the complexity of the interactions at play. The mutations associated with high viral loads can lead to drug resistance, posing a significant challenge to treatment efforts.

However, the study's findings on these mutations can be crucial in shaping future therapeutic strategies and the necessity for adaptive treatments. Exploring how gene expression varies in patients with persistent symptoms can illuminate mechanisms behind long-term COVID-19, potentially guiding long-term care approaches. Influencing Factors: Age, sex, comorbidities, and blood type can significantly impact the host response to infection. Studying these variables alongside gene expression can help identify at-risk populations and tailor preventive measures. Understanding the interplay between viral dynamics and host responses is essential for unravelling the pathogenesis of COVID-19, leading to better management and prevention strategies. The interactions between gene expression, viral load, and host factors directly influence morbidity and mortality- Elevated viral loads have been associated with more severe disease and higher mortality rates. Understanding the underlying gene expression patterns can help identify why some individuals experience worse outcomes. Immune response variability and variations in gene expression can affect how effectively an individual's immune system responds to the virus. This variability can influence both the severity of the disease and the likelihood of long-term complications. Comorbid Conditions: Individuals with pre-existing health conditions may have altered gene expression profiles that affect their response to infection, leading to increased morbidity and mortality. Long COVID Implications: The mechanisms driving long COVID, which include persistent symptoms and potential organ damage, can also be linked to gene expression changes, contributing to ongoing health issues in survivors and affecting overall morbidity in the population.

However, a more diverse patient cohort would enhance the generalizability of the findings across different populations, which is vital for understanding global implications. Recommended areas of improvement include longitudinal observations extending the follow-up period, which could provide insights into the long-term effects of gene expression changes on immunity and disease progression. Including a robust control group would strengthen the ability to isolate the specific impacts of SARS-CoV-2 on gene expression, adding rigour to the findings. In addition, it improves functional analysis, and a deeper exploration of how observed gene expression changes affect immune responses and disease outcomes would enhance the study's practical applications with critical evaluation. Discussing the research and its limitations, including potential biases and confounding factors, would provide a more balanced perspective on the findings. Further, replication in larger cohort groups replicating the study in more extensive, independent cohorts would validate.

6. PLOS authors have the option to publish the peer review history of their article (what does this mean?). If published, this will include your full peer review and any attached files.

Reviewer #1: **Yes: **Arnaud John KOMBE KOMBE

Reviewer #2: **Yes: **MONISHA KANDALA

---

## [Author Response · Author response to Decision Letter 0]

9 Oct 2024

We appreciate the opportunity to respond to the reviewers and to re-format the manuscript. We thank the reviewers for their comments and we believe we have responded appropriately to their comments.

We have removed the funding statements from the manuscript, however we would like to acknowledge that this work was supported by two NIH grants: CA125123 (CJC) and U19AI144297 (VA, CJC, RG, JP, PAP). Please change our financial disclosure to reflect this.

Please see below our response to the comments raised by the reviewers.

Sincerely,

Pedro Piedra

Chad Criegton

Reviewer 1: We thank the reviewer for the positive feedback.

Reviewer 2: We thank the reviewer for the positive feedback and the comment about adding limitations of our work. We have now added a paragraph in the Discussion stating the limitation of the study and the generalizability of our results.

---

## [Decision Letter · Decision Letter 1]

20 Dec 2024

Longitudinal host transcriptional responses to SARS-CoV-2 infection in adults with extremely high viral load

PONE-D-24-25457R1

Dear Dr. Avadhanula,

We’re pleased to inform you that your manuscript has been judged scientifically suitable for publication and will be formally accepted for publication once it meets all outstanding technical requirements.

Kind regards,

Mahendra Kumar Verma

Academic Editor

PLOS ONE

Additional Editor Comments (optional):

Reviewers' comments:

Reviewer's Responses to Questions

**Comments to the Author**

1. If the authors have adequately addressed your comments raised in a previous round of review and you feel that this manuscript is now acceptable for publication, you may indicate that here to bypass the “Comments to the Author” section, enter your conflict of interest statement in the “Confidential to Editor” section, and submit your "Accept" recommendation.

Reviewer #1: All comments have been addressed

Reviewer #2: All comments have been addressed

2. Is the manuscript technically sound, and do the data support the conclusions?

Reviewer #1: Yes

Reviewer #2: Yes

3. Has the statistical analysis been performed appropriately and rigorously? 

Reviewer #1: Yes

Reviewer #2: Yes

4. Have the authors made all data underlying the findings in their manuscript fully available?

Reviewer #1: Yes

Reviewer #2: Yes

5. Is the manuscript presented in an intelligible fashion and written in standard English?

Reviewer #1: Yes

Reviewer #2: Yes

6. Review Comments to the Author

Reviewer #1: the authors have adequately addressed the comments raised in a previous round of review and the new version of the manuscript is now acceptable for publication

Reviewer #2: (No Response)

7. PLOS authors have the option to publish the peer review history of their article (what does this mean?). If published, this will include your full peer review and any attached files.

Reviewer #1: **Yes: **Arnaud John KOMBE KOMBE

Reviewer #2: **Yes: **MONISHA KANDALA

---

## [Editor Report · Acceptance letter]

8 Jan 2025

PONE-D-24-25457R1 

PLOS ONE

Dear Dr. Avadhanula, 

I'm pleased to inform you that your manuscript has been deemed suitable for publication in PLOS ONE. Congratulations! Your manuscript is now being handed over to our production team.

Kind regards, 

on behalf of

Dr. Mahendra Kumar Verma 

Academic Editor

PLOS ONE